# Perceived Physical Literacy Is Associated with Cardiorespiratory Fitness, Body Composition and Physical Activity Levels in Secondary School Students

**DOI:** 10.3390/children10040712

**Published:** 2023-04-12

**Authors:** Charlie Nezondet, Joseph Gandrieau, Philippe Nguyen, Gautier Zunquin

**Affiliations:** 1Laboratoire Mouvement, Equilibre, Performance, Santé (MEPS), Université de Pau et des Pays de l’Adour, Campus Montaury, 64600 Anglet, France; 2L’unité de Recherche Pluridisciplinaire Sport, Santé, Société (URePSSS), Université de Lille, 59000 Lille, France; 3Laboratoire Motricité Humaine Expertise Sport Santé, (LAMHESS), UPR 6312, 06000 Nice, France; 4Departement “Unité Transversale des Activités Physiques pour la Santé” (UTAPS), Centre Hospitalier de la Côte Basque (CHCB), 64100 Bayonne, France

**Keywords:** exercise, public health, weight status, overweight adolescents, fitness

## Abstract

Background: Overweight and obese adolescents are often associated with lower Physical Activity (PA) levels and low Cardiorespiratory Fitness (CRF). Recently, the concept of Physical Literacy (PL), has been suggested to be associated with higher levels of active behavior and better health in adolescents. The purpose of this study is to investigate the relationships between PL, body composition, cardiorespiratory fitness and physical activity levels in French secondary school students. Methods: The level of PL was assessed in 85 French adolescents using a French version of the Perceived Physical Literacy Instrument (F-PPLI). Cardiorespiratory fitness was measured by the “20 m adapted walk/shuttle run test”. The PA level was assessed by the Youth Risk Behavior Surveillance System questionnaire. Weight status was measured by the Body Mass Index (BMI) and the body composition data. Results: We find a significant association between the PL and the percentage Fat Mass (%FM) (r = −0.43; *p* ≤ 0.01), between the PL and moderate to vigorous PA (MVPA) per week (r = 0.38; *p* ≤ 0.01). The PL was associated (r = 0.36; *p* ≤ 0.01) with the percentage of Skeletal Muscle Mass (%SMM) and cardiorespiratory fitness (r = 0.40; *p* ≤ 0.05). Conclusions: Developing the PL for the most disadvantaged secondary school students in a PA program could be a suitable strategy to increase their PA level, reduce their adiposity, and promote better long-term health.

## 1. Introduction

In children and adolescents, overweight and obesity are identified by the International Obesity Task Force’s (IOTF) Body Mass Index (BMI) cut-offs. Overweight is defined as a BMI between the IOTF cut-offs of 25 and 30 and obesity as a BMI above the IOTF cut-off of 30 [1]. The international prevalence of pediatric overweight and obesity for adolescents (10 to 19 years) is 24.9% of which 7.1% are obese [2]. In France, the prevalence of overweight (including obesity) is stabilizing but remains high, with respectively 18% of girls and 15% of boys being overweight and 3.8% of girls and 4.2% of boys being obese [2]. Being overweight (including obesity) during adolescence is independently a risk factor for developing obesity and its associated metabolic complications in adulthood [3].

On the other hand, recent national and international epidemiological studies confirm an increasing trend in time spent in behaviors described as “obesogenic” (e.g., physical inactivity, increasing time spent in sedentary behaviors and increasing consumption of sugar-sweetened beverages...) [3,4]. In France in 2020, only 34% of boys and 20% of girls aged 11–14 years meet the World Health Organization (WHO) recommendations of 60 min of Moderate to Vigorous Physical Activity (MVPA) per day [5]. This percentage dropped by 8% for girls and 2% for boys between 2014 and 2018. The French prevalence of physical inactivity is like the international prevalence, where 80% of youth under the age of 18 are considered inactive [6]. This decrease in levels of Physical Activity (PA) is accompanied by a significant decrease in Cardiorespiratory Fitness (CRF) which is the first component of physical fitness [7] and physical skills of adolescents (endurance, strength, speed, and agility) [8]. The decrease in levels of PA and CRF are detrimental factors for the metabolic, cardiovascular, and psychological health status of adolescents [9,10].

The interdependence of PA levels with physical fitness is important. Indeed, Hui et al. [11] showed that adolescents who met their daily PA guidelines were more likely to be more physically fit (Cooper Institute, 2017), compared to adolescents who did not [12]. In adolescent girls, the relationship between the MVPA and weight status (BMI Z-score and percent body fat) has also been shown to be significantly mediated by CRF [13]. Thus, the highest levels of CRF are found in adolescents with the highest PA levels. Conversely, adolescents with the lowest CRF levels are those with the lowest PA levels [14]. Given that the levels of PA are a determinant of adolescent health, the adolescents most at risk of developing cardio-metabolic risks are those with low CRF and low PA levels [13,15].

MVPA and CRF are also negatively correlated with weight status and the body composition of adolescents [16,17]. Overweight and obese adolescents, then, have a lower MVPA and physical fitness than other adolescents [16]. There is a negative correlation between the percentage of Fat Mass (%FM), PA levels, and the walking distance performed during a six-minute test (6MWT) in adolescents aged 11 to 14 years [16,18]. There is also a positive relationship between the percentage of Skeletal Muscle Mass (%SMM) and CRF (mean age 14.4 years (±2.5)) [19].

Thus, adolescents with low levels of CRF and low levels of PA associated with high BMI, high %FM, and low %SMM are most at risk of developing cardiometabolic complications in adulthood. However, we can note that, independently of weight status, improving CRF in adolescents would reduce these risks [20,21]. It is, therefore, a priority to develop complex, multi-modal strategies during childhood and adolescence to acquire and maintain active behaviors throughout life [22].

Recently, the concept of Physical Literacy (PL) has gained increased attention and has become a key element in the issue of promoting PA [23]. It can be defined as “the motivation, confidence, physical competence, knowledge and understanding to value and take responsibility for engagement in physical activities for life” (International Physical Literacy Association, 2014) [24]. PL enables people to possess different physical skills but also cognitive-affective skills (knowledge, understanding...) that allow them to adhere to PA throughout their lives and, thus, prolong individual health in the long term [25].

PL is a key component of an active lifestyle and «optimal» health. This concept was first explored in the field of education (particularly physical and sports education). Later, Cairney et al. [26] proposed an initial conceptual model linking PL, PA, and health. Early empirical evidence recognizes PL as a process for developing and increasing the PA levels in adolescents. Brown et al. [18] found positive associations between PL levels and participating in PA and between the PL levels and CRF [27]. An individual with a high PL level would be more likely to achieve the PA recommendations according to Kanellopoulou et al. [28]. Recently, Caldwell et al. [29] have shown that a high level of PL is a determinant of an adolescent’s health (low body fat percentage, better recovery from progressive aerobic tests (lower post-exertional heart rate), lower resting systolic blood pressure, and better quality of life). There is a negative association between PL and BMI in children aged 10–12 years [28]. On the other hand, this relationship is also found between %FM and PL [30]. Developing the adolescent’s PL is therefore essential and could be an interesting strategy to increase the levels of PA and, in the long term, benefit the health of adolescents, especially those with a high-risk profile (high adiposity, low fitness, and low PA levels) [26].

To our knowledge, we are the first study to examine the relationship between PL, body composition, PA levels, and CFR in high school students. Moreover, in France, no correlational study has been carried out on the theme of PL and health in adolescents. Therefore, we are interested in this.

A high level of PL among French adolescents could, therefore, be associated with high levels of PA and a superior CRF, determinants of long-term health.

The aim of this study is, therefore, to evaluate and investigate the relationships between PL, body composition, CRF, and the levels of PA in a population of French adolescents.

## 2. Materials and Methods

### 2.1. Procedure

Four 6th-grade classes (“Marracq” Secondary school in the city of Bayonne, France) took part in this study. These four classes were chosen because they volunteered to participate in the study.

The evaluations took place between February and June 2021. For each class of approximately 30 students, the assessments took place during physical education and sports classes over four specific sessions distributed over four consecutive weeks. The order of the assessments was the same for all classes: (1) 20 m shuttle run/walk test (TMNA-20), (2) Youth Risk Behavior Surveillance System, (3) Perceived Physical Literacy Instrument (PPLI), and (4) Anthropometric data and impedance measurement. The first author of the publication and the physical education and sports teachers supervised these evaluations.

Parents signed a consent form agreeing to data collection for the project and all subjects gave their informed consent for inclusion before they participated in the study.

The study was conducted in accordance with the Declaration of Helsinki, and the protocol was approved by the Research Ethics Committee of the Sciences and Techniques of Physical and Sports Activities (CER STAPS) (n°2020-11-02-44).

### 2.2. Participants

A total of 84 participants were needed to reach a power of 0.8 with an effect size of 0.3 and an alpha of 0.05 (*z* Fisher test). The inclusion criteria for our study were the agreement of physical education, for the sports teachers to participate in the study of their class, that the teenagers are in 6^th^-grade classes, and that they can participate correctly in all the evaluations. Any limitations in the adolescents that made it impossible to complete the assessments were a non-inclusion criterion.

A total of 85 adolescents, including 32 girls and 53 boys with an average age of 12.1 (±0.4) years, participated in our study. In total, 72 of them were normal weight, 2 were underweight, and 11 were overweight and obese.

### 2.3. Anthropometric Data

The height was measured to the nearest 0.5 cm using a wall height gauge (Seca^®^, 22089, Hambourg, Germany) according to the standard procedure: adolescents stood with their feet together, without shoes, and leaned against a wall with their heads, shoulders and feet aligned.

The weight was measured to the nearest 0.5 kg with a balance (Terraillon^®^, model Pop One; China). The weighing took place in minimal clothing (t-shirt, shorts, and socks). The BMI was then calculated using the formula: BMI = weight (kg)/height (m)^2^.

Underweight, overweight, and obese have been classified according to age and gender-specific BMI cut-offs [31,32].

### 2.4. Impedance Measurement

Body composition was measured with the multifrequency bioelectric impedance (Biody XpertZM II, AMINOGRAM SAS, la Ciotat, France). All measurements were recorded during the same morning. No vigorous to intense PA was performed 12 h before the test. Adolescents had an empty bladder and did not consume alcohol or caffeine-based beverages at least 24 h before the test. The measurement protocol was standardized and explained to everyone. The adolescent, in a seated position, placed his right hand on the rear sensors of the bioelectrical impedance and placed the front sensors at the level of the rear foot under the ankle.

The raw results (rate (kg), percentage of FM (%FM), and percentage SMM (%SMM were then provided by the “BiodyManager” (version 1) software.

### 2.5. Measurement of CRF by the 20 m Shuttle Walk/Run Test (TMNA-20)

The CRF was assessed by the CRF on the adapted 20 m shuttle run/walk test (TMNA-20) [33]. This test is an adapted version of the “Multistage 20-m shuttle run test” [34]. The aim of this test is to run continuously on a track with two blocks at each end, 20 m apart. The adolescent begins the test by walking at a speed of 4 km/h, then every minute the speed increases by 0.5 km/h until the adolescent stops voluntarily. The test is interrupted at the adolescent’s request or by the educator if the adolescent is no longer able to keep to the speed requested by the test tape. When the test is stopped, the Maximum Aerobic Speed reached (VMA) by the adolescent is estimated. The cardiorespiratory condition is then estimated as a correlation between the VMA (km/h) and the aerobic capacity (mL·kg·min^−1^). A mathematical formula is then used: (19.66 + (2.21 ∗ VMA) + (0.05 ∗ age) + (2.08 ∗ girl (0) or boy (1)) − (0.38 ∗ BMI)).

### 2.6. Youth Risk Behavior Surveillance System (YRBSS)

The levels of PA of secondary school students were assessed by the French adaptation of the self-administered Youth Risk Behavior Surveillance System (YRBSS) questionnaire [35,36]. This questionnaire provides a subjective estimate of PA levels with the calculation of the MVPA minutes per week of secondary school students. The questions concern the number of days spent in high-intensity physical activities for at least 20 min (D1), medium-intensity PA for at least 30 min (D2), and the number of days and duration of Physical Education and Sports classes (PE). To interpret this questionnaire, and calculate the minutes of MVPA per week, a mathematical formula is used: (2.5 ∗ D1 ∗ 20 min) + (D2 ∗ 30 min) + (PE ∗ 30 min). As described in “l’étude nationale nutrition santé ENNS, 2006” [37], this indicator allows us to classify adolescents into three categories: ^1^ <150 min of MVPA per week; ^2^ between 150 and 210 min of MVPA per week; ^3^ ≤210 min of MVPA per week.

With these results, it is possible to calculate the minutes of MVPA per week, which is an indicator of PA levels. 

### 2.7. Perceived Physical Literacy Instrument (F-PPLI)

The level of Perceived Physical Literacy (PPL) was assessed using the test from Sum et al. [38]: Perceived Physical Literacy Instrument (PPLI). The PPLI was validated in 2018 to measure the PPL in adolescents [25]. The PPLI was then translated into a French version (F-PPLI) by Gandrieau et al. [39] and tested in young adults (18–25). The F-PPLI is composed of nine items that are themselves divided into three dimensions: three items for the dimension “knowledge and understanding”, three items for “Sense of self and self-confidence” and three items for “self-expression and communication with others”. Each item is assessed using a 5-point Likert scale (1 = strongly disagree and 5 = strongly agree). The total PPL score is represented by the sum of the scores for each item. The total score ranges from 0 to 45 points (0 being very poor PPL and 45 being very good PPL).

### 2.8. Statistical Analysis

All statistical analyses are carried out by Statistica software (version 7.1).

The mean, standard deviation, median, and distribution of the data are calculated for each variable: age, BMI, %FM, %SMM, maximum aerobic speed (Vmax), aerobic capacity, PPL score (points), and PA levels (MVPA per week)). The normality of each variable was tested using the Shapiro-Wilk test. Due to the non-normality distribution, the non-parametric Mann-Whitney U test is used to test the gender-dependent variations in each variable. A *p*-value ≤ 0.05 was chosen for statistical significance.

Simple linear regression models are used to determine the relationships between PPL (independent variable) and the different dependent variables: BMI, %FM, %SMM, aerobic capacity (mL·kg·min^−1^), and MVPA (min/week). Using the Pearson correlation analysis, the correlation coefficient (r) is used to determine the strength of the relationship and the significance is indicated by a *p* value ≤ 0.05.

The mode of association between the variables is indicated by the standardized beta coefficient (β) and its 95% confidence interval.

Following on from the results of the Caldwell et al. [29] study, mediation analyses are conducted to determine whether the relationships between the PPL and the different dependent variables are mediated by MVPA (min/week).

For each relationship between the PPL (independent variable) and a Dependent Variable (DV), a multiple linear regression is used to determine the mediation of the MVPA variable (Mediator Variable).

## 3. Results

This is a sample of 85 secondary school students in the 6th grade who took part in the study. All students carried out the assessments. The detailed characteristics can be found in Table 1.

There was no significant difference in the age variable between girls and boys (*p* = 0.33).

### 3.1. Differences between Girls and Boys on Body Composition

Girls have a higher BMI and %FM than boys (*p* < 0.01), and boys have a higher %SMM than girls (*p* < 0.01).

### 3.2. Gender Differences in CRF and MVPA

The results of the fitness test show a higher maximal aerobic speed and maximal oxygen volume in boys (*p* < 0.01). Boys also had a higher MVPA per week than girls, both in (328.53 min (±152.80) vs. 232.69 min (±99.78); *p* = 0.01). For the PPL scores, girls did not score higher than boys (*p* = 0.05).

### 3.3. Associations, between PPL and Dependent Variables (BMI, %FM, %SMM, CRF, and MVPA)

Table 2 shows the results of the regression modelling. The results show a significant negative association between the PL and the %FM (r = −0.43; B = −0.86 (−1.30; −0.42); *p* < 0.01) (Table 2). The PL was also significantly positively correlated with the %SMM (r = 0.36; B = 0.46 (0.17; 0.75); *p* < 0.01). There was no significant association between PL score and BMI (r = −0.19; B = −0.11 (−0.24; 0.02); *p* = 0.096).

We found a significant positive association between the PL score and CRF with aerobic capacity as an indicator (r = 0.40; B = 0.33 (0.13; 0.53); *p* ≤ 0.05). We also found significant positive associations between the PL score and the MVPA per week (r = 0.38; B = 13.97 (5.88; 22.06); *p* ≤ 0.01).

We observed associations between the dependent variables. There were negative associations between the CRF (aerobic capacity) and the BMI (r = −0.59; *p* < 0.01) and between the CRF and the %FM (r = −0.60; *p* ≤ 0.01). A positive association exists between the CRF (aerobic capacity) and the %SMM (r = 0.86; *p* ≤ 0.01). The CRF was also positively and significantly associated with the MVPA (r = 0.49; *p* ≤ 0.01). The MVPA was negatively and significantly associated with the %FM (r = −0.37; *p* ≤ 0.05) and positively and significantly associated with the %SMM (r = 0.51; *p* ≤ 0.01).

### 3.4. Mediation Analyses

The above simple linear regression analyses show that the PPL is directly associated with all dependent variables except for the BMI. The MVPA is then investigated as a mediator of the associations between the PPL and the %FM, the %SMM, and the aerobic capacity.

The variable MVPA does not mediate the relationship between the PPL and the %FM. Indeed, in the simple linear regression model, PPL is inversely and significantly associated with %FM (B = −0.86 (−1.30; −0.42); *p* ≤ 0.01). Testing the MVPA variable as a mediator variable, we also find an inverse and significant association between the PPL and the %FM (B = −0.68 (−1.28; −0.08); *p* = 0.05).

The simple linear regression model shows a positive and significant relationship between the PPL and the %SMM (B = 0.46 (0.17; 0.75); *p* ≤ 0.01). In the multiple linear regression model testing the MVPA variable as a mediating variable, the effect of PPL on the %SMM is no longer significant (B = 0.14 (−0.22; 0.49); *p* = 0.38). This means that the effect found in the simple linear regression is due to the mediation of the MVPA variable. A similar result was found in the relationship between the PPL and the CRF. In the simple linear regression model, the PPL was positively and significantly associated with the CRF (B = 0.33 (0.13; 0.53); *p* ≤ 0.05). By integrating the MVPA mediator variable, we no longer find a significant relationship between the PPL and the aerobic capacity (B = 0.08 (−0.15; 0.30); *p* = 0.49). The MVPA variable is therefore a mediator of the relationship between the PPL and the aerobic capacity.

## 4. Discussion

In this cross-sectional study, we sought to explore the relationships between the PPL, body composition, CRF, and PA levels in French secondary school students in the 6th grade. None of these relationships had been established before in France, especially among adolescents. In accordance with our hypothesis, we found that the higher the PPL score, the higher the CRF of adolescents (r = 0.40; *p* ≤ 0.05). As a mediator, MVPA per week can partly explain the effect of this relationship, as MVPA is directly associated with the level of PPL (r = 0; 38; *p* ≤ 0.01). The linear regression model showed that there was an inverse relationship between the %FM and the PPL score (r = −0.43; *p* ≤ 0.01). Secondary school students with a high PL score had a higher %SMM than those with a low PL score (r = 0.36; *p* ≤ 0.01)). This relationship between the PPL and the %SMM could be explained by the effect of MVPA.

### 4.1. Relationship between PPL et CRF

The development of PL allows for the engagement in lifelong PA through improvements in motor skills and cognitive-emotional competencies [40]. CRF is integrated with the notion of “motor skills” found in the definition and assessment of PL [26]. The direct relationship between PL and CRF has not been studied in France. Our study is, therefore, the first to observe a positive relationship between these two variables in French adolescents (r = 0.40; *p* ≤ 0.05). Thus, for each additional point on the PPL score, aerobic capacity increases by 0.33 mL·kg·min^−1^. The influence of the PPL on the CRF of secondary school students is partly explained by the effect of MVPA per week (measured in minutes per week).

A Canadian study of 249 children aged 10 years showed similar results with a positive relationship between the PPL (score) and the CRF (heart rate) [18]. This relationship was also mediated by daily physical activity levels (min/day). In contrast to our study, the level of CRF was assessed by the peak heart rate value during a Bruce protocol. Lang et al. [27] showed similar results in 9393 Canadian adolescents aged 8–12 years. Adolescents with CRF classified as high had better PL scores compared to adolescents with lower CRF. In this paper, CRF was assessed with a field test identical to our study, however, PL was assessed with the CAPL tool.

### 4.2. Relationship between PPL and PA Levels

Our results also show a positive association between the PPL scores and MVPA (r = 0.38; *p* ≤ 0.01). Thus, for each additional point of PPL, the MVPA increases by 13.97 (5.88; 22.06) minutes per week. This same relationship, an increase of 3.19 (2.00, 4.40) minutes per day, was found in the 2020 study by Caldwell et al. in Canadian adolescents [29]. Brown et al. [18] showed that students with high PL profiles were the most involved in PA compared to students with moderate or low PL profiles. The same result was found between students with a moderate PL compared to those with a low PL [18]. Thanks to this longitudinal study, the follow-up of the results shows maintenance of these associations between the PL profile and the PA profile during the three years of follow-up. Thus, the development of PL is an important way to increase the PA levels and improve CRF [26].

Inactive and sedentary adolescents with low CRF are the adolescents most at risk of developing overweight and obesity problems [41,42]. Genetic factors combined with an obesogenic environment may explain the occurrence of overweight and obesity in inactive and low CRF adolescents. These same findings are found in our study. Adolescents with poor CRF have a higher BMI and adiposity and a lower %SMM.

### 4.3. Relationship between PPL and Body Composition

Our results also showed relationships between the level of PPL and the body composition of adolescents. There was a negative association between the PPL and the %FM (r = −0.43; *p* ≤ 0.01). For each additional point on the PPL score, the %FM decreased by 0.86%. This association is direct and not mediated by MVPA. This negative relationship between the PPL score and the %FM was also found in 249 adolescents. In this study, the assessment of PL was performed using the PLAY tools [43]. In the study by Caldwell et al. [29], the relationship between PL and body fat percentage was explained by the effect of MVPA.

Mendoza-Muñoz et al. [30] found a negative relationship (r = −0.49) between body fat and the total PL score studied with the CAPL-2 in 72 children. In contrast to our study, the sample had a much higher proportion of overweight and obese children (38.5% vs. 12.9%).

In our study, PL is also positively and moderately associated with the %SMM (r = 0.36; *p* ≤ 0.01)). It is positively associated with CRF in adolescents and negatively associated with mortality in adults [19,44]. Thus, for each additional point on the PL score, the %SMM would increase by 0.46%. In contrast to the %FM, the increase in the PPL score, allowing the increase in the %SMM, is explained by the MVPA.

### 4.4. Relationship between PPL and BMI

Our results show no significant association between the BMI and the PPL score. This lack of association could be due to the presence of our sample of normal-weighted individuals (84.7%). On the other hand, the BMI has many limitations in its interpretation and in particular the non-differentiation between FM and lean mass. BMI is only an index of body shape, so we can assume that the perception of the level of PL is more strongly impacted by the MVPA and by the %SMM and the %FM than by the BMI. In contrast to our study, a negative correlation was found between the BMI and the total PL score in Spanish children [30]. In a systematic review analyzing the relationship between health literacy and BMI, 5 out of 22 articles studied showed that children and adolescents (3–19 years) with low levels of health literacy have a higher BMI [45]. Kanellopoulou et al. [28] found an inverse relationship between health literacy scores and body weight in 1728 Greek adolescents (10–12 years), particularly among girls.

Thus, the adolescents most at risk are those with a high %FM and a low %SMM with the lowest levels of PA and CRF. Adolescents with these characteristics are those with low levels of PPL. The PPL was associated with body composition data, PA levels, and CRF, and some of these associations were mediated by MVPA. Improving the PL level of at-risk secondary school students could therefore increase their PA levels and, ultimately, improve their health.

### 4.5. Strengths and Limitations of the Study

Our study is the first French study to assess these parameters and to show associations between the PPL, CRF, MVPA, and body composition of secondary school students. Until now, studies looking at PL and its associations with health indicators were in North America [18,28,46]. The explanatory factors of the associations between PPL, body composition, and CRF remain to be investigated, but our study shows that MVPA could be the first explanatory factor in the evolution of these relationships.

Another strength of our study is the rigor in the choice of methodological tools used to assess the CRF and weekly reported MVPAs. The 20 m adapted Shuttle Walk/Run test (TMNA-20) [33] and the Youth Risk Behavior Surveillance System (YRBSS) [35,36] are validated and widely used for this population. The use of %SMM and %FM as primary indicators is a strength. They enlighten us on the links between body composition and PPL. Most studies showing an association between PL and anthropometric data have used BMI alone [47]. BMI is only an indicator of body composition and not a physiological index of health and, therefore, has only a partial impact on perceptions of health status and PA levels.

On the other hand, this experimentation is the starting point for a study on the longitudinal follow-up of overweight and obese adolescents. The objective of this project (CAPACITES 64) will be to develop the PL of adolescents to increase their PA levels and improve their long-term health. As PA levels are predictors of associations between PPL, body composition, and CRF in adolescents, it is important to promote increased PA levels and participation in PA.

This study has some limitations: The first is that the adolescent sample was not random. This type of sample presents a possible risk of recruitment bias.

On the other hand, a larger sample size would have allowed us to extend the results to the entire adolescent population. Another limitation is that we did not take into account biological parameters such as pubertal development and growth. These parameters directly influence the CRF of adolescents but do not influence the level of PPL [48,49].

The subjective assessment of the levels of PA may have a methodological limitation on the accuracy of the results. Indeed, unlike an accelerometer, the questionnaire does not assess the amount of MVPA in an objective way. Adolescents could, therefore, have increased or decreased their PA levels in an involuntary manner. The final limitation concerns the subjective assessment of PL. This assessment was carried out by a self-administered questionnaire. The results are therefore dependent on a PPL score. The fact that this variable is assessed in a perceived and self-reported manner may lead to variations and inaccuracies in the results as adolescents may have difficulties assessing their health behavior.

## 5. Conclusions

This study shows that in French secondary school students, the level of PPL is positively associated with CRF, MVPA, and the %SMM and negatively associated with the %FM. These results confirm our hypothesis that a good level of PL is associated with good CRF, high levels of PA, and positive health markers. This study confirms the fact that PL is a key parameter in the promotion of PA and health among French secondary school students. The results are preliminary but encouraging, as the development of PL in these secondary school students could help them maintain an active lifestyle.

## Figures and Tables

**Table 1 children-10-00712-t001:** Characteristics of participants (n = 85).

Characteristics	Total Sample	Girls	Boys	
	Mean (standard deviation)/median	Mean (standard deviation)/median	Mean (standard deviation)/median	*p*-value
n	85	32 (38%)	53 (62%)	
Standard-weighted	72	26	46	
Underweight	2	0	2	
Overweight	8	5	3	
Obesity	3	1	2	
Age (years)	12.1 (±0.4)/12.1	12.1 (±0.4)/12.12	12.1(±0.4)/12.1	0.33
Weight (kg)	41.3 (±8.4)/39.8	44.8 (±7.4)/44.9	39.2 (±8.3)/37.3	<0.01
Height (m)	1.5 (±0.07)/1.48	1.5 (±0.1)/1.5	1.5 (±0.1)/1.5	0.02
Body Mass Index (BMI) (kg/m^2^)	18.5 (±2.6)/17.8	19.5 (±2.4)/19.0	17.9 (±2.6)/17.2	<0.01
Fat Mass (FM) (%)	14.1 (±8.9)/13.4	18.1 (±9.4)/18.5	10.5 (±6.8)/9.2	<0.01
Skeletal muscle mass (SMM) (%)	45.6 (±5.7)/46.0	40.70 (±3.3)/41.1	49.9 (±3.4)/50.1	<0.01
Maximum aerobic speed (VMA) (km/h)	11.2 (±1.4)/11.5	10.2 (±1.0)/10.0	11.8 (±1.2)/12.0	<0.01
Aerobic capacity (mL·kg·min^−1^)	39.3 (±4.3)/39.7	35.3 (±2.7)/34.8	41.7 (±3.0)/42.3	<0.01
Perceived Physical Literacy Score (PPL)	37.9 (±5.4)/39.0	38.8 (±4.7)/40.0	37.2 (±5.7)/39.0	0.05
MVPA (min/week)	328.5 (±152.8)/320.0	232.7 (±99.8)/230.0	372.1 (±153.4)/390.0	0.013

BMI: body mass index; FM: fat mass; SMM: skeletal muscle mass; VMA: Maximum aerobic speed; PPL: Perceived Physical literacy; MVPA: Moderate to vigorous physical activity (minutes/week); *p*-value: probability of rejecting the null hypothesis (*p* = 0.05) calculated by the Mann–Whitney *U*-test (non-parametric variables). Total PPL score is 1 to 45 points.

**Table 2 children-10-00712-t002:** Associations, between PLL and the dependent variables: BMI, %FM, SMM, aerobic capacity, and MVPA per week.

Variables	Β (95% IC)	r	*p*-Value
BMI Perceived Physical Literacy Score	−0.11 (−0.24; 0.02)	−0.19	0.096
%FM	−0.86 (−1.30; −0.42)	−0.43	≤0.01
%SMM	0.46 (0.17; 0.75)	0.36	≤0.01
Aerobic capacity (mL·kg·min^−1^)	0.33 (0.13; 0.53)	0.40	≤0.05
MVPA (min/week)	13.97 (5.88; 22.06)	0.38	≤0.01

BMI: body mass index; FM: fat mass; SMM: skeletal muscle mass; MVPA: Moderate to vigorous physical activity (minutes/week); B: regression coefficient; CI: confidence interval; r: correlation coefficient; *p*-value: probability of rejecting the null hypothesis (*p* = 0.05).

## Data Availability

The datasets used during the current study are available from the corresponding author on reasonable request.

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
