# Peer review of "Perceived Physical Literacy Is Associated with Cardiorespiratory Fitness, Body Composition and Physical Activity Levels in Secondary School Students"

_children, 2023, doi:10.3390/children10040712_

Round 1

Reviewer 1 Report

Line 35- include parameters for determination of overweight and obese people

Line 110- exclude statement " we are therefore dealing with convenience sample" because maybe it is appropriate for logistics reasons "convenience sample" but has it shortcomings in terms of methodology

124- how they reach 0.8 statistical power ? Could you describe in more detail sample size methodology 

Line 131- exclude statement- irrelevant for this section of research

Result section- could you  highlight  if it  is normal distribution or not? You use a non-parametric analysis for all variables in table 1.

Line 336-  there is a large amount of literature which refers to this issue among Europe for example, there is no need to highlight specific part of world

Reviewer 2 Report

The topic of this manuscript is more interesting, and I am also more interested in this research topic. Physical literacy does contribute to the adolescents’ health, so the research on this topic has not only theoretical implications, but the value of practical application as well. After reading this manuscript, I mainly put forward two suggestions of improvement for your reference.

l  Issues on Statistics

1.      According to the title of your manuscript, I think you set wrong independent variables and dependent variable. I think the independent variables should be perceived physical literacy, cardiorespiratory fitness, body composition, and the dependent variable should be physical activity levels. 

2.      It is recommended to use multiple linear regression instead of binary linear regression, because R2 can only be used to explain the proportion of the dependent variable explained by the regression equation, and cannot explain the correlation between the independent variable (X) and the dependent variable (Y). Therefore, I suggest you change the independent variable to BMI, % fat mass, % skeletal muscle mass, aerobic capacity(ml/kg/min-1) and physcial literacy, and change the dependent variable to MVPA(min/week). The degree of influence from each independent variable on the dependent variable (MVPA) can be learned by establishing a multiple linear regression equation. 

3.      If the multiple linear regression equation can not established or the equation is not ideal, it can be changed to Pearson correlation analysis to compare the strength of the correlation between each independent variable and MVPA. 

l  Issues on Writing

1.      Please change £210 minutes of MVPA per week to ³210 minutes of MVPA per week (Line 177-178);

2.      Please change LP to PL (Line 98 and Line 404);

3.      Please change all “and al.” in the manuscript to “et al.”, for example: Hui et al. (Line 51-52), Cairney et al. (Line 83), etc.

Reviewer 3 Report

The Article “Relationship between perceived physical literacy, cardiorespiratory fitness, body composition and physical activity levels in secondary school students interesting but, in my opinion, not very well written scientific article. The major strength of the study is this study practical application, and it is relevant for real situation. Nevertheless, it requires several changes before it will be published. There are some remarks concerning this article:

1.     My biggest concern is related to the research presentation. In my opinion the research is done with such little number of students.

2.     In Introduction part there is so many places 1-2 sentences presented as separate paragraphs. Somehow it seems the not linked to other parts (e.g., lines 64-97).

3.     Additionally, I would like to note that in introduction part is a to many stretch on just French population. Without any remarks on general PA, PF and PL tendencies in the world.  

4.     Another issue related to research methodology and procedure. Authors declare that they did research in February and June 2021. I would like to get clarification how it was possible to do research in this period while it was 3rd lockdown in France, and it was all constraints related to Covid-19 pandemic.

5.     Moreover, in the lines 108-110 presented information about in the study involved schools. It is not clear what information there is presented in the brackets, especially numbers.

6.     Bioelectrical impedance method description in different places presented differently (e.g. lines 144-145 – “hand-foot”, and in lines 149-150 – “right hand”.

7.     There are some citation inaccuracies (e.g., 206-207 lines).

8.     Tables presented to wide (e.g., table 2).

9.     In results and discussion part appears some abbreviations and it is nowhere explained (e.g. %SMM, %FM).

10.  Reference list for such kind of scientific article is appropriate, cited 45, but could be expanded, because some of references are too old, 17 references published not recently, especially it is noticeable in discussion part. Some of references presented not correctly, not in accordance with all requirements, without DOI etc. (e.g., 1, 32 etc.). Some references have not been presented in results and discussion part (e.g., 54).

Round 2

Reviewer 1 Report

Authors made significant improvements and the paper is suitable for publishing

Reviewer 3 Report

 Authors of this publication “Perceived physical literacy is associated with, cardiorespiratory fitness, body composition and physical activity levels in secondary school students has taken into account most of my remarks made during my firsts review process.

1     1.      In my first review round I have mentioned that in my opinion authors to many focused on just French population. Additionally, I would like to ask or suggest at least to note, that in the World and in France trends on physical activity and obesity are similar or not.

2    2.       Additionally, I suggest adding to the article more recently published data for the comparison in discussion part.
